# Using Wash’Em to Design Handwashing Programmes for Crisis-Affected Populations in Zimbabwe: A Process Evaluation

**DOI:** 10.3390/ijerph21030260

**Published:** 2024-02-23

**Authors:** Astrid Hasund Thorseth, Jennifer Lamb, Constancia Vimbayi Mavodza, Mandikudza Tembo, Anesu Petra Mushangari, Eddington Zinyandu, Thomas Heath, Sian White

**Affiliations:** 1London School of Hygiene and Tropical Medicine, Keppel Street, London WC1E 7HT, UK; 2Biomedical Research and Training Institute, 10 Seagrave Road, Avondale, Harare, Zimbabwe; 3Action contre la Faim, 21 Giraffe Crescent, Borrowdale West, Harare, Zimbabwe; 4Action contre la Faim, 102 rue de Paris, CS 10007, CEDEX, 93558 Montreuil, France

**Keywords:** handwashing, programme design, behaviour change, hygiene, humanitarian crises, process evaluation

## Abstract

Wash’Em is a process that supports humanitarians in assessing and designing rapid but context-specific hygiene programmes in crises or outbreaks. The process consists of training implementers, using tools to learn from populations, and entering findings into a software which generates contextualised activities. A process evaluation of Wash’Em use was conducted in a drought-affected area in Midland province, Zimbabwe. Data were collected during the programme design and following implementation using a mix of qualitative methods. Findings were classified against the intended stages of Wash’Em, and the evaluation domains were defined by the UKRI Medical Research Council. The Wash’Em process was not fully implemented as intended. An abridged training was utilised, some of the tools for learning from populations were omitted, many of the recommended activities were not implemented, the delivery modalities were different from intended, the budget available was minimal, and the number of people exposed to activities were fewer than hoped. Despite these ‘on the ground’ challenges and adaptations, the Wash’Em process was considered feasible by implementers and was seen to be less top–down than most programme design approaches. The populations exposed to the intervention found the activities engaging, understood the content, and reportedly took action to improve handwashing behaviour. Programmes such as Wash’Em, which facilitate community participation and are underpinned by theory and evidence, are likely to yield positive results even if processes are followed imperfectly.

## 1. Introduction

### 1.1. Hygiene Programming in Crises

Humanitarian emergencies such as natural disasters, disease outbreaks, or armed conflicts cause displacement of populations, the destruction of social systems and infrastructure, and present increased public health risks. These conditions create the ideal environment for the spread of communicable diseases [1,2]. Of particular concern are diseases which transmit through a faecal–oral route. Faecal–oral pathogens include diarrhoeal diseases, some respiratory infections, and many outbreak-related diseases (e.g., cholera) and are a leading cause of preventable illness and death across all types of humanitarian crises [1]. Handwashing with soap is known to be one of the most cost-effective public health interventions and can result in diarrhoeal disease reductions by up to 48% [3,4] and reductions of respiratory infections of by up to 23% [5,6,7]. In stable settings (locations not affected by crises), handwashing promotion should be facilitated by exploring what determines whether people wash their hands. Handwashing interventions which have been developed based on an understanding of these determinants have been proven to change behaviour [8,9]. Handwashing promotion in humanitarian crises typically utilises hygiene education (e.g., communicating the health benefits of handwashing) and the provision of hygiene products (e.g., provision of soap and handwashing facilities). This narrow focus on handwashing knowledge or infrastructure has proved insufficient to change handwashing behaviour in these settings [10]. This could be because these represent just two of the many possible determinants influencing handwashing practices of crisis-affected populations.

The limitations of hygiene programming in humanitarian crises have been recognised [11,12] but change within the humanitarian sector has been slow. Prior research [12,13,14] has identified that this is likely to be because current behaviour change approaches have not been designed with humanitarian contexts in mind and are often explained in long, written documents that are hard to apply within humanitarian timelines. Secondly, Water, Sanitation, and Hygiene (WASH) practitioners see behaviour change as something outside their remit and the competency of development actors. Hygiene programmes are often under-funded and either replicate activities which practitioners have tried in other contexts or rely on external consultants to develop more tailored programming. Lastly, even though implementers are keen to improve their programmes based on evidence, they often struggle to find the time and capacity to contextualise and adapt ideas that have worked elsewhere [12,13,14].

### 1.2. The Wash’Em Process

Wash’Em is a process designed to help implementers to assess and design rapid, evidence-based, and context-specific hygiene behaviour change programmes. The Wash’Em process is intended to make designing and implementing behaviour change programmes more feasible for implementers, irrespective of their prior training or experience, thereby mitigating the need for external ‘experts’ to be flown in to support programming. The first step to using the Wash’Em process is to learn about what the process involves (Figure 1). Organisations can pick from a range of training formats based on their needs, including a facilitated online course and a face-to-face training for implementers [15]. The second step in the Wash’Em process is for implementing staff to use a set of five Rapid Assessment Tools to learn about the determinants of handwashing behaviour from crisis-affected populations in their setting. The Rapid Assessments are participatory methods which focus on the determinants of handwashing behaviour that are most likely to be affected in a crisis and are designed to generate the kinds of data needed to influence program design. The five Rapid Assessment Tools are described in Table 1 and can be viewed via Appendix A. The third step is to summarise the findings from the Rapid Assessments by entering them into the Wash’Em software [16] and answering 48 multiple choice questions. This prompts the software to select from more than 80 recommended handwashing activities, those which are most likely to change behaviour based on the contextual determinants. Each activity comes with a step-by-step guide to aid organisations in planning the logistics and delivery of their programme. The Wash’Em process was designed based on several years of research and iterative improvements based on feedback from humanitarian actors [14,17,18]. It can be completed in as little as two days [19] and has already been used in more than 94 humanitarian responses since March 2020.

Although the Wash’Em process has been widely used by implementing actors, this uptake has happened largely independently, with humanitarians in crisis-affected settings discovering the tool and using it without consultation with the Wash’Em developers. This has meant that it has been challenging to understand the ‘on the ground’ successes and challenges that are being faced by implementing partners, how the process is being adapted to suit different contexts, and whether the Wash’Em designed activities are acceptable and relevant to populations. This process evaluation is designed to track ‘on the ground’ experiences with implementing each phase of Wash’Em in a crisis-affected setting in Zimbabwe. This is done with a view to understanding contextual pathways of change in this setting, while also contributing to the broader aim of understanding whether Wash’Em improves the process for developing acceptable, feasible, and context-appropriate hand hygiene programmes in crisis-affected settings.

## 2. Materials and Methods

### 2.1. Study Site and Population Demographics

The study took place in two districts of the Midlands Province in Zimbabwe. Most of the population in this region earn their living through agricultural activities, with the main produce being cotton [21]. Zimbabwe has experienced a prolonged water and food security crisis in recent years due to increasingly severe economic challenges, rapidly rising inflation, and climate hazards [22,23]. This insecurity was exacerbated by the COVID-19 pandemic [21]. A baseline survey conducted by Action Contre La Faim/Action Against Hunger (ACF) and Africa Head (AA) in January 2022 covered the two study districts and found that 40% of respondents used surface water as their main source of drinking while 36% used boreholes. More than 50% of respondents reported to spend more than 30 min travelling to and from the water point and 72% of the respondents said they did not have access to adequate water. Only 15% of households had a dedicated place to wash hands with soap [24] compared with 64% nationwide [25,26].

ACF, in collaboration with local partner AA were funded to implement a programme entitled ‘Community System Strengthening for Reducing Vulnerability, Restoring Economic Sustainability, and Improving Recovery from COVID-19 in Zimbabwe’ in the study districts from July 2021 to January 2023. This multipronged humanitarian initiative included a component on WASH. ACF and AA planned to use Wash’Em to design the handwashing promotion component of the programme. In addition to hygiene, AA intended to repair and rehabilitate 48 boreholes which were dysfunctional and drill an additional 6 boreholes. ACF and AA planned to deliver their programme, including the handwashing component, in partnership with the local government and Village Health Workers (VHWs). Village Health Workers are unpaid volunteers that work under the Environmental Health Technicians (EHTs) to help promote public health initiatives at a community level in Zimbabwe. AHA and AA also intended the Wash’Em designed activities to be delivered to communities through their Community Health Club (CHC) model [27,28,29,30] which brings together community members on a weekly basis to learn about and tackle health challenges in their community. This approach, they envisioned, would help to ensure that the Wash’Em activities reached most of the population, were linked to other WASH initiatives and led to sustained action.

### 2.2. Study Design and Framework

A theory of change was developed to provide a framework for this study and outline how the Wash’Em process intended to influence programme design. This was informed by the UKRI Medical Research Council (MRC) Guidelines for Process Evaluations of complex interventions [31] which is widely used for the evaluation of public health interventions. This is presented in Figure 2 in Section 3. This study was designed to assess four process domains which related to the steps that the organisations (AA, ACF, local government counterparts, and the communities) followed to design the programme following the Wash’Em process. It also assessed nine programme domains relating to the actual programme that was implemented following the use of the Wash’Em process. All 11 domains were informed by the process evaluation guidance developed by Linnan and Steckler [32], the MRC Guidelines and the stages of implementation as defined in in the Wash’Em programme design process. The domain definitions are available in Table 2.

### 2.3. Data Collection

Data were collected from the start of the implementation of the Wash’Em designed hygiene programme (August 2022) and for the subsequent 6 months. A mix of qualitative methods were used including interviews with implementing staff, observations, photography, and note taking throughout the design process and implementation; focus group discussions (FGDs) with the targeted crisis-affected populations; and secondary analysis of operational documents and programme reports. FGDs, interviews, and observations were conducted by experienced qualitative researchers (CVM and MT) who trained one assistant to support data collection. Table 3 provides a summary of the methods used, their intention and the sample size used.

### 2.4. Interviews with Wash’Em Implementers

Interview participants were purposely sampled from all staff that were involved in the Wash’Em training or implementation of Wash’Em designed activities. Sampling was designed to include a mix of genders (because prior research has indicated that the gender of programme or evaluation staff can have a bearing on outcomes) [8], experience, and positions within the implementation team. The in-depth interviews followed an interview guide (S6 Document) and aimed to investigate fidelity, context, acceptability, and feasibility of the Wash’Em process and programme. Interviews with these staff members took place after the Wash’Em implementation. In-depth interviews were conducted in person or remotely via Zoom, depending on the location and availability of staff. Interviews were performed in either English or Shona language—whichever the staff member was more comfortable using. Interviews were led primarily by staff from Biomedical Research and Training Institute (BRTI) in Zimbabwe who were fully external to the implementation process. Staff from London School of Hygiene Tropical Medicine also supported interviews of AA and ACF staff remotely via zoom.

### 2.5. Observation, Notetaking, and Photography

Observation was used to assess whether the Wash’Em process was implemented as intended. The observation focused on key moments of the Wash’Em programme delivery including select moments during the delivery of Wash’Em designed activities. All observations were recorded on semi-structured observation forms which were specifically designed to track the intended steps at each implementation stage. Staff also took free-form notes and photos to complement this process. Observation was conducted by staff from BRTI.

### 2.6. FGDs with Crisis-Affected Populations

FGDs were held with crisis-affected populations living in villages where the Wash’Em designed handwashing programme was implemented. The study team recruited participants that could recall attending a meeting about handwashing in the last three months. All interviews were led by Shona speaking facilitators from BRTI. The focus group discussions followed an FGD guide (S7 Document) and aimed to investigate the acceptability and relevance of Wash’Em designed activities, participant engagement, and response to the activities, and contextual factors affecting hygiene. FGDs took place at two time points, 2 and 8 months after the end of the implementation of Wash’Em designed activities. The second round of FGDs, conducted in May 2023, was performed because, after a preliminary analysis of the data, the research team concluded that saturation had not been reached with the initial sample. The second round of interviews allowed us to reach data saturation, providing fuller perspectives and enhancing confidence in the themes identified.

### 2.7. Secondary Analysis of Programme Documents

The implementing organisations provided several documents for secondary analysis. These included findings from the Rapid Assessments, outputs from the Wash’Em software (Version 2, Centre for Affordable Water and Sanitation Technology, Calgary, AB, Canada), the broader programme baseline report, and the programme plans and budgets.

### 2.8. Data Management and Analysis

All interviews and FGDs were audio recorded, transcribed, and translated into English. The interview data were analysed thematically following the process outlined by Braun and Clarke [33] which included (1) familiarisation with the data; (2) development of an initial deductive coding tree based on the 11 process and programme domains described in Table 2; (3) generating themes with the aid of NVivo 12 software (QSR International, Cambridge, MA); (4) validating themes by ensuring that both coders (AHT and CVM) discussed disagreements and identified common patterns; (5) defining themes; and (6), summarising and visualising these themes in relation to the process and program domains and the postulated theory of change. Observational data, notes, photos, and videos were discussed by the evaluation team (LSHTM, BRTI and monitoring staff from ACF) and compared to the intended stages of Wash’Em use and the intended implementation of Wash’Em designed activities. Programmatic documents were also reviewed by the evaluation team. Through discussion, the evaluation team came to a consensus understanding about the degree of fidelity and adaptation made during implementation. This allowed links to be made between the data in the programmatic documents and the qualitative interview and observation data also collected as part of the process evaluation.

## 3. Results

### 3.1. Description of Study Participants

In-depth interviews with implementers were conducted with 11 staff (one from ACF, four from AA, four EHTs, and two VHWs). Three were female and eight were male. Those interviewed had between 1 and 13 years of experience working on either WASH programmes or on community health promotion. Nine focus groups were organised with 6–15 adult male and female participants in each. In total 56 people participated: 22 men and 34 women.

### 3.2. Category 1a: Implementation of the Wash’Em Process for Programme Design Compared to the Intended Process

#### Fidelity and Coverage

Figure 2 uses a traffic light system to indicate the extent to which the intended Wash’Em process was followed by in country implementers, with green indicating that the activity was completed as intended, orange indicating that the activity was partially carried out, and red indicating that the activity was not carried out or carried out substantially different from what was intended. This visual summary and the written summary below are derived from observations of implementation and interviews with the implementation staff.

AA staff were introduced to the Wash’Em approach by ACF, and two AA staff were invited to attend a global online training on Wash’Em, enabling them to develop a deep understanding of the approach and what it was designed to do. Wash’Em was written into an AA funding proposal, which expressly indicated that Wash’Em would be used to design the hygiene promotion component of their programme. However, the allocated funding covered the cost of the trainings, but no money was left for implementation of the Wash’Em designed programme. Staff within AA were excited to try a new approach to handwashing behaviour change because they recognised the limitations of some of their past programming. AA staff were introduced to the standard Wash’Em training materials [16] during the global online training and then decided to modify and contextualise these materials to suit their purposes and allow them to deliver the training in a shorter space of time (1 day compared to the recommended 3 days). Ultimately, they delivered two in-person training sessions to a total of 22 staff across the two districts. Trainees included other staff from AA, Environmental Health Technicians (EHTs) from local Government, and some volunteer Village Health Workers (VHWs). The condensed training timeline meant that some of the recommended training modules were covered rapidly or in some cases not covered at all. As such some implementation staff reported gaps in their understanding of Wash’Em.

The trained staff were able to travel to the programme sites the day after the training to use the Rapid Assessments. These were pre-translated by the AA staff but were not piloted with populations in this context prior to use (as per the Wash’Em recommended process) due to limited time and access to the programme locations. Data collection in both districts was complicated by the fact that the population was relatively dispersed, the districts were difficult to traverse, and the teams were only able to allocate 7 h to data collection in each site. Due to these tight time constraints only three of the Rapid Assessment were utilised, with Disease Perception and Personal Histories being omitted in both districts because AA staff felt they were less relevant to their context (Table 4, Quote 1).

Of the three Rapid Assessments that were implemented, the Touchpoint and Motives tools were implemented as intended within six focus groups. Twenty people participated in the Handwashing Demonstrations, but implementers reported prioritising houses nearby (therefore not applying guidelines for selecting a diverse sample) (S1 Document) due to time limitations and the dispersion of the population.

After collecting the data, staff in District 1 collectively entered the findings into the decision-making tables and Wash’Em programme designer software. Due to power outages in District 2, this collective process was not possible and instead was performed by the lead trainer. In both districts the Handwashing Demonstrations Tool indicated that there were no handwashing facilities present at the household level. Soap was available in half of the households but reported to be kept inside the main house, away from the kitchen and toilet. Water was also not stored in a location where it could be conveniently accessed for handwashing. In both districts, the Motives Tool revealed a need to increase the perceived link between handwashing and nurture (being a good parent) and disgust avoidance (being seen as a person who is neat). In District 1 handwashing was also linked to the motive of attractiveness (being seen as an attractive person) while in District 2 it was linked with status (being seen as a wise and well-educated person). The Touchpoint Tool indicated that the most effective ways of reaching the population in District 1 were through radio content, messaging at public transport hubs, or through community meetings and events. In District 2 more people indicated that they had access to mobile phones and the tool also identified that schools and religious institutions could be appropriate ways of reaching the population.

Once the recommendations had been generated by the Wash’Em software, the training facilitators discussed the suggested activities with the training participants and decided which to implement. A total of six activities were recommended by the software for each district, with only one activity (‘The Power of Soap’) being recommended in both sites. A summary of the activities is presented in Table 5 and a full description of each activity is available in SM3. In District 1, all activities were taken forward but, in District 2, only one activity was taken forward and the rest of the programme utilised the activities recommended for District 1. This was because implementation staff felt that the finding from the Touchpoints Tool, which indicated that most of the population had mobile phone access, was not totally reliable as often people do not have coverage, credit, or power to charge phones. Therefore, any activities that utilised phones or social media were dropped. AA also explained that they had a strong preference to continue to deliver hygiene programming through community events as that is what they are accustomed to. The ‘Watching Eyes’ activity was dropped because this was difficult to install with the type of handwashing facilities (Tippy Taps) that they were promoting. Once the activities had been decided on, many of the subsequent aspects of project planning (work plan development, procurement, and staff training) were performed by engaging key staff as needed.

### 3.3. Category 1b: Implementation of the Wash’Em Designed Programme Compared to the Intended Process

#### 3.3.1. Fidelity

The Wash’Em designed activities were intended to be implemented alongside a renewed curriculum for CHCs. However, due to delays in implementing the Wash’Em process, this was not possible and instead the previously designed activities were implemented by VHWs as they went house to house or worked with small groups doing hygiene promotion. As well as handwashing, the VHWs promoted menstrual hygiene management, the construction and safe use of household sanitation facilities, COVID-19 prevention behaviours, waste management, food hygiene, and community water supply management.

Most Wash’Em implementation events were held outside community centres in the shade, under trees. All consumables required for the activities were purchased ahead of time by AA staff members. Due to the lack of funds available for implementation, staff had to draw from core budgets and other projects to cover the costs of the materials needed for implementation. Table 3 presents an overview of the activities that were implemented. All activities were implemented with some local adaptations, reported in receipt and change mechanisms (Table 4, Quote 2).

At each Wash’Em implementation event, 25 basic handwashing facilities and 25 bars of soap (a bucket with a tap attached often known as a veronica bucket) were distributed to the participants. Not all participants could receive a bucket as the budget only allowed for the purchase of 300 buckets. The distributions became divisive in some settings with participants feeling disappointed that they did not receive a handwashing facility when others did. In one community the VHW ensured there were only 25 people attending the implementation event. This approach led to a more peaceful and positive event but excluded members of the community from the programme. When asked about the organisation’s reasoning for purchasing and distributing a limited number of handwashing facilities, the implementers referenced the general tips section on the Wash’Em Programme Designer which states that, when working with small hygiene budgets, organisations should focus resources on improving handwashing facilities and making soap more available for their population. The recommendations say this should be prioritised over the ‘soft’ part of the hygiene promotion given that handwashing infrastructure is so key for enabling practice. In addition to distributing handwashing facilities the implementers demonstrated how Tippy Taps [34] can be constructed from locally sourced materials.

#### 3.3.2. Coverage, Dose Delivered and Received

Initially, Wash’Em was intended to be implemented in every district and ward covered by the wider ACF and AA WASH response programme. However, due to delays in getting ethical approval for this study, AA and ACF agreed to implement the traditional handwashing promotion programme from the CHC in most wards in the two districts. A smaller group of three wards from each district that were not included in the CHC implementation were chosen to receive a customised Wash’Em designed intervention, as described above.

Implementation of the Wash’Em designed activities was conducted 2–3 months after the completion of the Wash’Em process to allow time for procurement and planning. In District 1, six implementation events were conducted. In District 2, six implementation events were planned and four completed (Table 5). The number of participants at each event varied from 34 to 90. Most events had more women attending than men, due to men being busy at work during the day. In some instances, aging populations were unable to attend due to the challenge of transport to the event location. Observation notes indicated that the Wash’Em designed activities were delivered through stand-alone events on a range of weekdays. But in these districts, community meetings usually happen on Thursdays so leveraging these existing meetings could have led to higher attendance at implementation events (Table 4, Quote 3).

According to the implementors, the VHWs in one ward in District 2 did not carry out the implementation of the hygiene programme due to the disgruntlement of the VHWs due to a lack of incentives.

### 3.4. Category 2a: Receipt and Mechanisms of Change (Implementers)

#### 3.4.1. Feasibility (Process)

Overall, the senior implementers attending the training of trainers expressed that they found it useful and informative. The training left them prepared to organise and deliver their own face to face trainings. Reflecting on the structure of the implementer training, one staff member explained that, although it was feasible to complete the training modules in one day, it was very intense and did not allow time for the participants to pause and reflect on the individual modules, due to the amount of content that had to be covered.

Due to power cuts at the venue for the training in District 2, the venue had to be evacuated due to the high temperatures inside without air conditioning. The training then had to be moved outside and completed in the shade of a large tree and using only printed materials as laptops used for presenting slides quickly ran out of power. The training facilitators were also concerned about the number of materials that had to be printed as it was a time and budget consuming task. However, they reflected that all the printed materials were useful and necessary for the effective completion of the training and allowed the participants to keep information about the training to refresh their memory at a later stage. The costs of the two implementer trainings and data collection was 2552 USD, and included meals for training participants and venue costs.

When using the Rapid Assessment Tools, implementers appreciated that the Wash’Em process allowed time for consulting with members of the community before designing and implementing hygiene promotion programmes (Table 4, Quote 4).

Collecting data using the Rapid Assessment Tools was considered a laborious process by implementers due to the long travelling distances from the training venue to the villages where crisis-affected populations were living. Furthermore, the distances between each household were far and only accessible by foot. Once the data collection team was present in the village, they used the AA implementers devices to capture quality videos for the Handwashing demonstrations tool.

The implementers reported that according to the Wash’Em training of trainers’ curriculum, the Personal Histories Tool should be used during a crisis or an outbreak. While their proposal defined the settings as being affected by a prolonged water and food security crisis, the AA implementers did not see this as being equivocal to the kind of disaster referred to in the tool, with staff viewing the current drought as ‘not that bad’. Furthermore, implementing staff explained that they understood that the Disease Perception Tool should only be used during or after a disease outbreak (such as COVID-19 or cholera), and that the chronic, high rates of diarrhoea in the districts did not merit being considered as an outbreak or a disease of concern.

#### 3.4.2. Feasibility (Programme)

When ACF and AA’s wider WASH program was planned with a budget of 59,375 USD, there was no funds allocated for the implementation of Wash’Em designed activities. This meant that the AA staff had to request additional funds to cover the cost of implementation. A total of 2340 USD was spent on 300 handwashing facilities and 300 bars of soap. Other consumables needed for the implementation included food dye, bread, paper for printing and creating commitment cards, turmeric, and Vaseline. However, no clear financial record exists for this spending and therefore was not able to be recorded accurately (Table 4, quote 5).

The EHTs were paid their normal salary as they are employed by local government offices and the AA staff were on employment contracts and paid for their work. However, the implementers leading the implementation events and follow up, the VHWs, were not paid for their work and this was a significant barrier to their motivation for delivering Wash’Em activities. Some VHWs refused to implement activities due to lack of incentives, citing that other organisations would provide financial incentives for similar work. The VHWs, despite participating in the Wash’Em process, highlighted that they were overwhelmed as they are the focal entry point for all implementing agencies in the district. Given that AA did not provide financial incentives to the VHWs, they viewed the Wash’Em process and the associated activities as very intensive and time consuming. Ultimately, some VHWs reported prioritising the activities of the agencies that provided them with incentives, at the expense of Wash’Em.

### 3.5. Category 2b: Receipt and Change Mechanisms (Crisis-Affected Population)

#### 3.5.1. Acceptability and Relevance

The recipients valued the fact that the hygiene promotion events in which the Wash’Em designed activities were implemented, were short in duration. Sessions took between one to two hours.

The activity ‘Being pulled in all directions’ (Table 5) was implemented a total of 10 times in the two districts. AA officers made some adaptations when they translated the activity instructions, including adding local examples of chores a mother would usually have. The local adaptation allowed for a mother or a father to play the main role in the activity (Figure 3).

Participants in the FGDs had the intended realisation this activity was designed to create, that is, that after participating in this exercise they were able to see how women have a lot of chores that they need to do daily, and as such important behaviours like handwashing can sometimes be deprioritised (Table 4, Quote 6). Observers noted that crisis-affected populations were eager to attend and were actively engaged in the activity but, as the implementation events included both men and women, some women were felt uncomfortable engaging with men, given the physicality of this activity.

The ‘Dye on food’ activity (Table 5) was amended by implementers to not include the first part where a table of food was set. Instead, the activity started by setting up activities where hands could be contaminated such as going to the toilet or changing a baby’s nappy, then immediately handling food or feeding the baby, therefore leading to contamination and facilitating the spread of germs. In the FGD, participants noted how the ‘Dye on food’ activity really helped to visually demonstrate how germs could move from one contaminated area and spread into their household, which would explain the diarrhoea and cholera challenges that sometimes would be experienced in their area (Table 4, Quote 7) (Figure 4).

In the activity known as ‘Pledges’ (Table 5), the implementers instructed participants to build on their experience of the ‘Power of soap’ activity which had demonstrated the importance of washing hands with soap and asked, ‘What things can you agree on that will help your community wash their hands with soap or ash regularly?’. Participants then agreed on several commitments to make together as a community: (1) building suitable toilets, (2) digging trash pits, (3) keeping food covered, (4) investing in handwashing stations around the homestead, (5) being more active in personal hygiene, and (6) investing in buying soap as a community. In some cases, the VHW facilitators sought and received the support of the headman and chief of the village. Implementers felt these hierarchies were a powerful influence to support this activity. VHW perceived that Wash’Em activities would continue beyond the presence or facilitation of AA because of the headman’s involvement and commitment to what was written on these pledges.

The ‘Child life game’ (Table 5) was implemented four times in District 2. In one village, this activity helped combat a local belief that children’s teething was a primary cause of diarrhoea. After the interactive play, the participants discussed with the facilitators and agreed that it was more commonly the child’s contaminated hands or other items they put in their mouth to sooth their sore gums that was the source of germs causing diarrhoea, not the teething itself. The activity ‘Child life game’ (Table 5) was not implemented in District 1 as the implementers thought 5 handwashing activities were sufficient to promote handwashing in each village.

AA staff members prepared ‘Commitment cards’ (Table 5), which were printed on paper at their offices before distributing these to VHWs to implement the activity at household level. The AA facilitators encouraged VHWs to work with each family to come up with a set of commitments, including building a toilet, water treatment and storage, digging a waste disposal, keeping their home environment clean, as well as regularly washing hands with soap. Ultimately this meant that the activity was slightly less community led than intended.

#### 3.5.2. Participant Engagement and Response

The crisis-affected population was observed to be engaged, interactive, and overall pleased with the content of the hygiene promotion event (Table 4, Quote 8). Participants expressed to the facilitators that they appreciated that the event did not last more than two hours, allowing them to get on with their day. Positive peer pressure through sharing knowledge as well as experiencing and understanding the handwashing promotion activities together elevated the community’s value placement on handwashing behaviours and associated WASH components.

#### 3.5.3. Mediators

The Wash’Em designed activities appear to have had a positive influence on the number of households which built handwashing facilities. This is based on self-reports by crisis-affected populations who were exposed to the intervention who also described a marked change in their own behaviour and that of their neighbours (Table 4, Quote 9). Secondly, the EHTs collect data on handwashing indicators every quarter and reported that, in the villages in which the Wash’Em designed handwashing programme was implemented, there was more than a 90% increase in handwashing facilities at household level (Figure 5). The handwashing facilities distributed by AA make up 57% of the new handwashing facilities reported. Within the scope of our research, we were not able to independently verify these self-reports or EHT data, however, if accurate, this would indicate an increase from less than 1% of households having handwashing facilities to a coverage of more than 70% after the implementation of Wash’Em. Village 9 and Village 10 were the villages where the VHW did not implement any Wash’Em designed activities apart from distributing 25 veronica buckets in each village, and in these areas no additional handwashing facilities were reported 4 months after implementation. When comparing the villages that did not have Wash’Em designed activities implemented, the difference in number of new handwashing facilities built was marked, reported implementers.

### 3.6. Category 3: Context

The main external barrier to facilitating improved handwashing in these districts was identified by implementers and populations as being access to water (Table 4, Quote 10). In addition to prolonged drought, the program was implemented during the dry season in August, with rains expected between October 2022 and March 2023.

Although solidly constructed Tippy Taps can withstand general use by humans, another barrier to the effect of the programme was the destruction of the home-built handwashing stations by livestock and by children playing. Members of the community appreciated receiving soap from AA with the handwashing distribution but were worried about where the next bar of soap would come from as the costs of the soap was a barrier for purchase. Implementers noted that, if the rainy season started as scheduled, they expected an increase in use of the newly distributed buckets and Tippy Taps for the collection of rainwater but felt that the additional access to water would ultimately make handwashing easier and more convenient for the community.

## 4. Discussion

Our findings indicate that, in this context, the Wash’Em process was not fully implemented as intended; despite this, important conclusions were still able to be drawn related to implementing this program in a resource-scarce area. Of the 19 first-level outputs (as described in Figure 2), 11 were coded green, indicating that they had basically been implemented as intended. The remainder were only partially implemented (seven outputs), not implemented or differed substantially from what was intended (one output). Specifically, our findings indicate that, in Zimbabwe, an abridged training was utilised; two of the five Rapid Assessment Tools were considered less relevant and omitted; many of the recommended activities were not implemented (particularly in District 2), the delivery modalities were different to what was proposed by ACF and AA but were also different to what was recommended by the Wash’Em software; the budget available was utilised on the initial CHC implementation and the training, leaving minimal financial resources for the actual implementation; and the number of people exposed to activities was fewer than hoped.

Of the second-level outputs defined in Figure 2, five of the nine were achieved in Zimbabwe. Implementers in general felt the Wash’Em approach was feasible, and populations exposed to the intervention reported that the Wash’Em designed activities led them to prioritise handwashing, take action around handwashing (e.g., building handwashing facilities), believe the behaviour was normative, and helped to address misconceptions around handwashing and diseases that they held.

However, other secondary-level outputs along the theory of change were only partially met. For example, implementing staff from ACF and AA did not feel that all of the Wash’Em recommended activities were relevant to their setting. Primarily, this was because, in District 2, the Touchpoints tool indicated digital media and radio may be effective ways reach populations, but based on the implementation team’s experiences this was likely to be impractical. The team opted to go for a more familiar delivery modality (in-person interactions) and an approach that was more aligned across the two districts. This decision meant that some of the nuance of the contextualisation that Wash’Em can offer was lost, and some of the other secondary-level outputs were not realised. For example, the Wash’Em activities also struggled to make handwashing more convenient and socially rewarded, because the activities that related to these outputs were dropped. The Wash’Em process does allow for ‘replacement activities’ to be selected if implementers feel some of the activities are less relevant, but this feature was not utilised in Zimbabwe. It is not unexpected that implementers will customise and change the recommended activities to suit their own needs. In this case, the decision made the programme easier to roll out across the two districts and allowed some innovative approaches to be utilised within a familiar delivery modality. Supporting innovation uptake within the humanitarian sector has been documented to take time. One reason for this is because the chaotic nature of operating in a crisis tends to make actors risk averse, less prone to adopting innovative ideas, and more likely to rely on what is familiar [35,36,37]. Overcoming this requires a broad understanding of humanitarian decision-making processes and consultations with Wash’Em users to understand how support can be provided at this programme planning stage.

Some of the implementation team felt that the standard Wash’Em use timelines (approximately a week) was still too time consuming for their needs and for the constraints of accessing communities in their setting. The process may have seemed time consuming because most programmes previously designed in this context were developed by senior staff and with less active participation from the community. This type of ‘top–down’ programme design is not unusual for crisis response programmes. Prior research on hygiene programme design in these settings [12,13,14] has indicated that implementers often have to compromise on more ‘ideal’ processes of programme design due to the perceived imperative to act with urgency and the associated time pressures and stress that come with this. As such programmes tend to rely on the past expertise of managerial staff to make decisions since it is not always possible for organisations to set aside time to learn from communities [14,36,38]. Even when engagement with communities does take place, many programme teams struggle to use these data to contextualise programmes [14]. However, when community engagement is done well and when learning feeds back into programme implementation, research indicates that programmes are more accepted, relevant, trusted, and likely to lead to positive outcomes [39,40,41,42]. Indeed, in our study, implementers recognised that, even though Wash’Em took more time, it was the in-person qualitative data collection that the implementing team valued and which they felt led to more contextualised and holistic programmes. There has been a strong push in recent years for anthropology and qualitative science to be better utilised to support humanitarian and outbreak programming [43,44,45]. However, achieving this has been inherently challenging because the ‘humanitarian worldview’ is often epistemologically and methodologically at odds with the anthropological approach. This has led to qualitative science sometimes being seen by humanitarians as unscientific, unpragmatic, time consuming, and something that requires specialist expertise [43,46,47,48,49]. Wash’Em provides a semi-structured way for humanitarians, with limited experience in qualitative methods, to engage with communities and immediately use the results to influence programme design. As the name suggests, the Rapid Assessment Tools are not intended to be as ‘deep’ as traditional anthropological methods but may serve as a useful way of starting to strengthen qualitative capacities in the sector, something that has been acknowledged as weakness in past responses [14,49,50].

The process of implementing Wash’Em may has also seemed burdensome to implementers in Zimbabwe because budgets were so limited. It is not unusual for humanitarian organisations to be working with limited budgets for hygiene programming and this has been recognised as a chronic challenge in the sector [14,19,51,52]. The Zimbabwean team decided to prioritise the limited funds they had for handwashing infrastructure. This was consistent with Wash’Em guidelines, which recommend this because creating convenient and desirable handwashing facilities is likely to be the most influential determinant of handwashing behaviour [18,53]. However, with limited funds, they were not able to purchase enough facilities for the whole target population. This created challenges for implementing staff and was divisive among communities. It is important that programmes can facilitate infrastructural changes in an equitable and sustainable manner. Global guidance on equitable commodity distributions exists [54], however, achieving this is likely to require a two-pronged approach consisting of increased and sustained financing from humanitarian donors and more effective engagement with communities to allocate resources and leverage local knowledge and innovations.

Wash’Em was implemented within a broader programme designed by ACF and AA which was intended to be multi-sectoral and address a range of needs facing the affected communities, including improving water access. This is consistent with Wash’Em guidance, which emphasises that handwashing programming should not be seen as a stand-alone initiative. However, in practice the programme was not able to meet the scale of needs in the target areas. For example, there were an increased number of functioning water points in the region as a result of the programme, but most people exposed to the Wash’Em component of the programme did not experience meaningful improvements in water access. Therefore, populations in our study indicated that, despite generally liking the Wash’Em activities, their lack of water access was still a major barrier to regular handwashing practice. This finding acts as a reminder of the importance of designing humanitarian programmes that are holistic, cross-sectoral, and sustainable. However, with current humanitarian funding only meeting 50% of global needs, the challenge of achieving meaningful change at scale is immense. Greater collaboration with government partners and the private sector is likely to be required to close the gap [55].

A further challenge facing Wash’Em’s implementation in Zimbabwe was that the VHWs who were primarily responsible for delivering Wash’Em activities in communities were not renumerated for the additional time the Wash’Em activities took. While the payment of per diems or other forms of financial incentives can have a complex impact on consequences [56,57], humanitarian programmes must be cautious that their programme design choices and delivery modalities do not create additional burdens on frontline staff, such as VHWs, who are often undervalued, time and resource limited, and have to deal with a multitude of responsibilities while working under challenging conditions [58,59]. Reflecting on these power dynamics, and the relationships between the different levels of implementing staff, should be an important aspect of undertaking quality programming [60] and future implementation science research.

### 4.1. Implication of the Findings for Improving the Wash’Em Process

The Wash’Em process has been improved based on some of these observed deviations from the intended process. For example, it is now a requirement that users complete at least four Rapid Assessments in order to generate sufficient data for contextualisation of activities. A stark finding of this evaluation was that the protracted nature of the water and food security crisis in Zimbabwe was not viewed as a ‘crisis’ by frontline staff, nor did they view chronic diarrhoea challenges in the region as being a critical disease risk necessitating the prioritisation of handwashing behaviour change initiatives. Other research has identified that handwashing is often not ‘problematised’ by populations during crises, nor is it prioritised by humanitarians for a variety of reasons [14,61]. However, these variations in understanding have caused the global-level Wash’Em trainers to rethink the way fragility, crises, and outbreaks are described and has prompted the team to reflect on how Wash’Em can be used to support resilience building and crisis mitigation programming. Issues related to budgeting for Wash’Em were already a concern for the Wash’Em team and, subsequently, guidance is now available on how to effectively write Wash’Em into a proposal and develop and adequate budget [62]. Finally, the results of this process evaluation highlighted that the stage where users assess the recommended activities, select which to implement, and develop programme plans, is key. The Wash’Em developers have subsequently placed stronger emphasis on this stage of the process and have developed a programme planning tool to guide implementing actors through questions related to delivery modality, sustainability, cross-sectoral programming, logistics, and procurement.

### 4.2. Limitations

The transferability of findings from process evaluations can be difficult to interpret due to the very nature of the study design which focuses narrowly on a particular intervention, in a specific context and time [63,64,65]. For this work, we anticipate that the nuances of the implementation of Wash’Em in Zimbabwe were highly contextually dependent, but that there will be higher transferability of findings related to the experiences of the implementation team. This is supported by anecdotal evidence and surveys conducted by the Wash’Em developers [19].

Our ability to undertake a robust process evaluation was affected by a 9-month delay in gaining formal ethical approval for the study in Zimbabwe. This meant that the implementation of the Wash’Em process started before the process evaluation could commence and, as such, we had to rely on secondary programmatic data describing the Wash’Em training, Rapid Assessment Tool use, data entry, and programme planning. These was then complemented with retrospective reflections on these stages of the process via the interviews with implementers. This inability to collect primary data during these stages of Wash’Em use (as initially proposed) resulted in some gaps in our understanding of how the Wash’Em design process was followed. Compressed timelines also resulted in us having to drop a household before and after survey which was intended to collect data on exposure to the intervention, mediating factors, and behavioural outcomes (assessed through a proxy measure of whether handwashing facilities with soap and water were available). Unfortunately, this means that our understanding of intervention mediators and behavioural impact is self-reported and likely to be subject to social desirability bias. These are common challenges with evaluating handwashing behaviour change interventions [66,67,68,69,70]. The delays we experienced reflect a broader challenge of undertaking implementation science in fragile settings, which is that evaluations are often funded as stand-alone research activities which must align with separately funded ongoing programmes. Under this type of common funding modality, it is challenging not only to align timelines but also to find common motivation across donor, programme implementer, and researcher interests. Similar challenges have been reported by others undertaking programme evaluations in complex crises or outbreaks [71,72,73]. Mitigating such challenges could be possible if donors prioritise process evaluations within programme funding and encourage greater collaboration on such grants between academic partners and implementing actors.

## 5. Conclusions

Despite the real-world challenges of implementing Wash’Em amid the constraints of tight project timelines, limited funding, difficult terrain, and minimal changes to water infrastructure, the overall benefits of using Wash’Em to inform programme design appear to have been appreciated by implementers and populations alike. The prospect of utilising a novel process, such as Wash’Em, to aid program design may initially seem daunting to implementers, but our findings indicate that programmes informed by community participation and underpinned by theory and evidence are likely to yield positive results even if processes are followed imperfectly.

## Figures and Tables

**Figure 1 ijerph-21-00260-f001:**
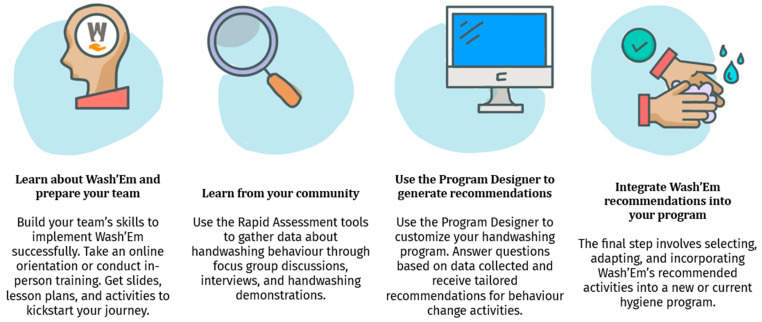
The Wash’Em process.

**Figure 2 ijerph-21-00260-f002:**
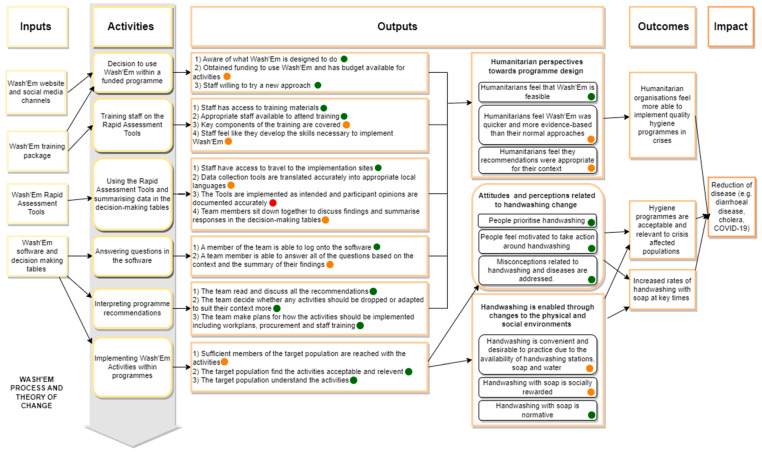
Wash’Em flow diagram depicting the phases of implementation and the corresponding key actions that implementing organisations need to take. A traffic light system indicating the extent to which the intended process was followed by implementers has been applied, with green indicating that the activity was completed as intended, orange indicating that the activity was partially carried out, and red indicating that the activity was not carried out or carried out substantially different from what was intended.

**Figure 3 ijerph-21-00260-f003:**
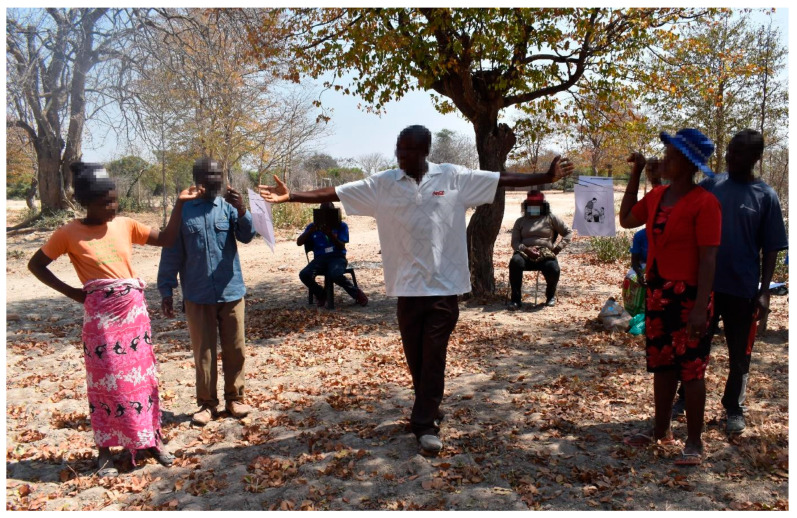
Implementation of the Wash’Em designed activity ‘Being pulled in all directions’.

**Figure 4 ijerph-21-00260-f004:**
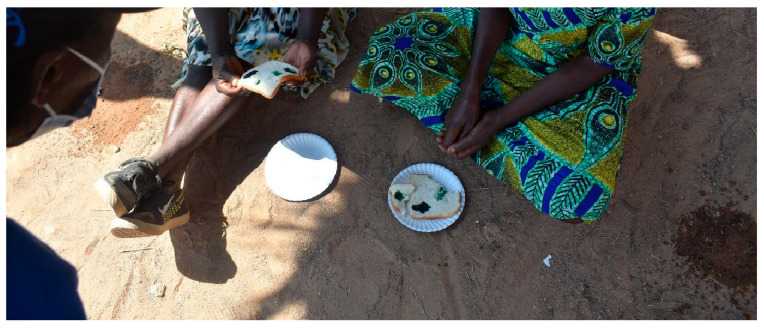
‘Dye on food’ was one of six Wash’Em designed activities implemented in District 1 and District 2.

**Figure 5 ijerph-21-00260-f005:**
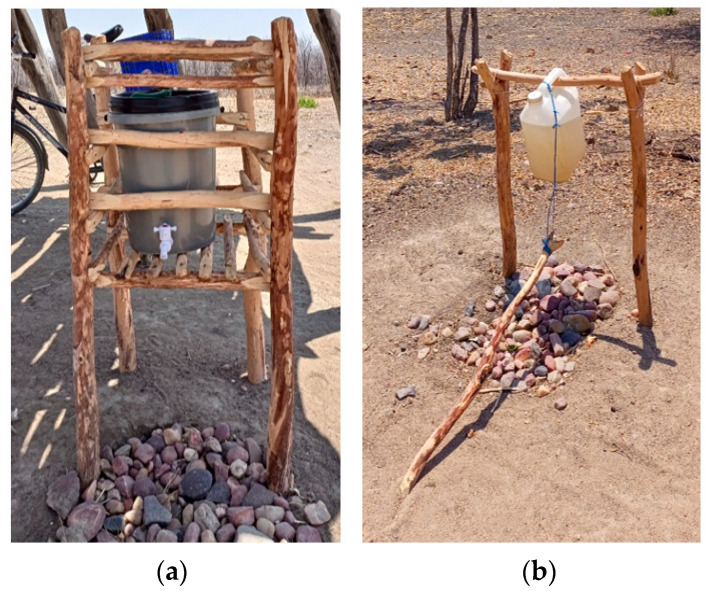
A handwashing facility observed in a village where Wash’Em was implemented. To the left (**a**) is one of the Veronica buckets that were distributed to 25 households in each village. On the right (**b**) is a Tippy Tap, constructed by another household in the same village after implementation of the Wash’Em designed handwashing behaviour change activities.

**Table 1 ijerph-21-00260-t001:** Description of the Wash’Em Rapid Assessment Tools [20].

Rapid Assessment Tool	Description	Modality and Suggested Sample Size	Further Resources
Handwashing Demonstration	Designed to generate quick insights into whether a person’s home and community environment enable or prevent handwashing practices. Participants are asked to demonstrate how they would normally wash their hands and data collectors’ video this process and then answer a series of questions about how participants interact with objects (for example, soap and containers) and infrastructure (for example, handwashing facilities and water points) and whether there are factors in the physical environment that may enable or create barriers to handwashing.	Individual method done at the household with a minimum of 10 people.	Full Rapid Assessment Guide available in Appendix A
Motives	Designed to identify what is driving handwashing behaviour or preventing it in a particular context and which motives shape people’s identity and other behaviours. Participants are introduced to a set of character cards that are linked to behavioural motives (e.g., an image of ‘A person who has lots of friends’ is used to epitomise the affiliate motive and the desire to belong to a social group). Participants are then asked to rank the character cards based on which character they think is most likely to always wash their hands with soap to the person who they think is least likely to practice handwashing with soap.	At least 2 focus group discussions with different sub-groups of the population.	Full Rapid Assessment Guide available in Appendix A
Disease Perception	Designed to understand people’s perceptions of the disease of interest (e.g., diarrhoea or cholera). The group is asked to identify 5 illnesses they are most worried about, the illnesses are then ranked in order of which ones they worry most about to which ones they worry least about. Participants are asked to grade their perceptions about the disease on Likert-style scales.	At least 2 focus group discussions with different sub-groups of the population.	Full Rapid Assessment Guide available in Appendix A
Personal Histories	Designed to obtain a broad understanding about the experiences of populations affected by crises or outbreaks. Participants are asked to talk about three different time periods (before, during, and after the crisis). Participants will draw an image that reflects how they looked and felt at each stage, to help participants open up and share their experiences with the implementers.	Individual method done at the household with a minimum of 6 people.	Full Rapid Assessment Guide available in Appendix A
Touchpoints	Designed to identify the types of delivery channels that are present in a context and prioritise which ones are likely to be most effective in reaching your population. Participants are presented with image cards depicting different touchpoints (e.g., radio, village meetings). Participants are then asked if each touchpoint reaches a lot of people in their community and identify sub-groups such as ‘women’ that this touchpoint is particularly effective in reaching.	At least 2 focus group discussions with different sub-groups of the population.	Full Rapid Assessment Guide available in Appendix A

**Table 2 ijerph-21-00260-t002:** Overview of the process evaluation categories and domains.

Category	Domain	Definition
1a. Implementation of the Wash’Em Process	Fidelity	The content and quality of the implemented Wash’Em design process compared with what was intended.
	Coverage	The degree to which staff participated in each stage of the Wash’Em process.
1b. Implementation of the Wash’Em Designed Programme	Fidelity	The content and quality of the implemented activities compared with the Wash’Em guidance for those activities.
	Coverage	The degree to which the crisis-affected population were exposed to the intervention.
	Dose delivered and received	The number of activities that were intended to occur as part of the Wash’Em implementation compared to what actually happened.
2a. Receipt and change mechanisms of the Wash’Em programme design process	Feasibility (Process)	The extent to which implementers feel they can follow the steps of Wash’Em and implement the Wash’Em designed activities in a crisis. Assessing this will take time, cost, logistics, and capacity into consideration.
	Feasibility (Programme)	The perceived feasibility of implementing Wash’Em designed activities according to implementing staff.
2b. Receipt and change mechanisms of the Wash’Em Designed Programme	Acceptability and Relevance	The extent to which crisis-affected populations feel the programme activities are acceptable, appropriate, and relevant to their needs and situation.
	Participant engagement and response	Receipt and understanding of key messages, and interaction with the programme content.
	Mediators	Specific behavioural determinants measured along the hypothesised causal pathway.
3. Context	Context	Anything external to the Wash’Em process that may have acted as a barrier or facilitator to its use for programme design, implementation, or its effects.

**Table 3 ijerph-21-00260-t003:** Overview of process evaluation methods.

Research Method or Data Source	Respondents	Purpose	Process Domains Covered	Programme Domains Covered	Sample Size
Interviews with Wash’Em implementers	Wash’Em Implementers	To understand the expectations of implementing staff in relation to the feasibility and usefulness of the Wash’Em process and its likely outcomes.	Fidelity Context Feasibility	Fidelity Context Acceptability	11 implementing staff
Observation, note taking and photography	Wash’Em implementers and to a lesser extent the crisis-affected populations that they are interacting with	To understand whether the Wash’Em implementation was implemented as intended and record whether any events deviate from the intended process.	Fidelity Context Coverage	Fidelity Context	12 implementing staff Observation to take place within the office and within implementation sites
Focus Group Discussions	Crisis-affected populations who are exposed to the intervention	To explore reactions to the Wash’Em activities and generate reflections on what was liked or disliked about them.		Acceptability Participant engagement and responses Context Relevance	9 focus group discussions
Secondary analysis of programmatic data	Wash’Em implementers	To gather data on planning, programme targeting, training, budgeting and programmatic adaptation.	Fidelity Context Feasibility Coverage	Dose delivered Fidelity Coverage Context	As available

**Table 4 ijerph-21-00260-t004:** Quotations from Wash’Em implementers and crisis-affected populations in a drought affected area of the Midlands Province, Zimbabwe.

Category	Domain	Quote	Quote Number
1a. Implementation of the Wash’Em Process	Fidelity	‘The disease perception tool wasn’t contextual because in the community where we worked no one was affected by COVID-19, no one was succumbed to any diarrhoeal disease. No one lost his or her relative. With regards to COVID-19, so we didn’t find it fit to conduct it but we did train the facilitators on the tool in case we have such a situation as a disease outbreak. We just used the three tools; handwashing demonstrations, motive mapping and touchpoints as they were the most relevant to the community.’ (Implementer, female).	1
1b. Implementation of the Wash’Em Designed Programme	Fidelity	About the commitment card activity: ‘Village Health Workers actually walked door-to-door and they explained to us what we were supposed to do. We wrote what we wanted to do and these were put on the doors or cupboards, we actually wrote timelines.’ (Community member, Female).	2
	Dose delivered and received	‘However, the challenge we faced when they came was that they conducted the door-to-door visit without informing us that they were coming so we had to come back from the farms to attend to them. Thus, there was lack of communication.’ (Community member, Male).	3
2a. Receipt and change mechanisms of the of the Wash’Em programme design process	Feasibility (Process)	‘I think, in terms of programming, I realised that most organisations usually they rely much on top–down approaches where interventions are not informed by community views and community perceptions with regard to their needs in hygiene promotion. So organisations just show up and they say this is the programme we have for you. And the community does not have any opportunity to share their views… The issue of collecting baseline data to inform programming, like we did using the Rapid Assessment Tools, it is a very good idea because it promotes informed programming by virtue of the communities also being involved in decision making on what needs to be done in their communities. I’ve seen that work very well.’ (Implementer, Male).	4
	Feasibility (Programme)	‘I’d say budgetary constraints, Wash’Em we are running on a shoestring budget. So that is a challenge.’ (Implementer, Female).	5
2b. Receipt and change mechanisms of the Wash’Em Designed Programme	Acceptability and Relevance	‘It [Being pulled in different directions] also zeroed in on the patriarchal nature of our society, were the women carry on the burden of the house chores. You know, and the men were like “aaah” [realised] our women are overwhelmed. So yeah, that was one of the impacts. Of course, the women who themselves are more hands on the day-to-day house chores appreciated that it was important to wash hands at every critical point, but the men were also in agreement that the women are overwhelmed.’ (Community member, Male).	6
		‘This [The dye on Food activity] was relevant because since this activity was done in our community it seems that children are not getting sick anymore. Children got sick very frequently in the past, before they came with this activity so it is likely that the mothers would feed their children without washing their hands.’ (Community member, Male).	7
	Participant engagement and response	‘I think this activity will influence our behaviour more this time because of the cholera outbreak everyone seems to be alert in terms of good hygiene practices. If we think about the amount of time spent doing different chores it shows that we have to pay serious attention in washing hands thus it will not affect us’. (Community member, Female).	8
	Mediators	‘This is something that we cannot forget because this activity was done last year, we still remember this information, and we are actually practicing it even if we are busy with other things.’ (Community member, Male).	9
3. Context	Context	‘Some of us, we have a challenge of water so if I use more water, then it means I should spend more time in fetching water to practice what we learnt. So for me I feel like AA could have given us a close source of water for us to practice this because we women suffer more in terms of fetching water.’ (Community member, Female).	10

**Table 5 ijerph-21-00260-t005:** Results of the Wash’Em Process for District 1 and District 2 of handwashing promotion activities recommended through the Wash’Em process. Full description of the activities is available in the Appendix A. “✓” means the activity was recommended and “X” means the activity was not recommended.

Activity	District 1		District 2	
	Activity Recommended?	No. Events Activity Was Implemented in	Activity Recommended?	No. Events Activity Was Implemented in
Being pulled in all directions: A participatory play about a hardworking mother designed to link handwashing to being a good parent.	✓	6	X	4
The power of soap: An interactive activity were people rub glitter on their hands and the try to remove it with water only, before successfully removing the glitter with soap. Designed to demonstrate that soap should always be used for handwashing.	✓	6	✓	0
Pledging: Community members and community leaders make public pledges to make handwashing a priority. Designed to make handwashing appear to be a normative and socially approved behaviour.	✓	6	X	4
Commitment card: A household-level planning tool to encourage people to take small doable actions towards improving handwashing behaviour. Designed to improve the sense of ownership around handwashing facilities.	✓	6	X	4
Dye on food: Using food dye, this activity is designed to demonstrate how germs can easily spread from hands to food.	✓	6	X	4
Child life game: A participatory play at a community event to show how the lives of two children can be dramatically changed by small moments of their lives, including the frequent practice of handwashing with soap.	✓	0	X	4
Social media tips: By creating a social media page or group where you can obtain tips from community members who have improved their handwashing facilities. Designed to help make handwashing seem normative.	X	0	✓	0
Testimonies from survivors: Document stories of disease survivors and their experiences of contracting a disease and then share this with other people in the community via video or radio. Designed to help population appreciate the full range of consequences if a family member gets sick.	X	0	✓	0
Can you smell the truth? Use a blindfold test to demonstrate how hands washed with soap smell good, while hands ‘washed’ without using soap do not. Designed to make participants realise that soap is key, and handwashing with water only is not effective.	X	0	✓	0
Don’t miss out on the experience: This activity involves creating a sign or mural in your community which highlights how great people feel after handwashing with soap. Designed to increase the association between handwashing and feeling comfortable.	X	0	✓	0
Watching eyes: Create stickers with a picture of eyes on them and place the stickers above handwashing facilities. Designed to make people feel like others are noticing whether they wash their hands with soap.	X	0	✓	0

## Data Availability

The data presented in this study are available on request from the corresponding author. The data are not publicly available due to the privacy of interview subjects.

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
