# Peer review of "Using Wash’Em to Design Handwashing Programmes for Crisis-Affected Populations in Zimbabwe: A Process Evaluation"

_ijerph, 2024, doi:10.3390/ijerph21030260_

Round 1

Reviewer 1 Report

Comments and Suggestions for Authors

There are very few studies available from Zimbabwe and authors should be commended for taking up this study.  Handwashing is  an important behavior during humanitarian crises which is the focus of this study. The article describes the qualitative process evaluation of Wash'Em handwashing program.

-- The introduction has been written well and is succinct.

-- The design is good. It talks about coverage, fidelity, acceptability, context, etc. However, in a process evaluation quality control is an important component which the chosen process evaluation model does not fully take into account.  Perhaps this should be mentioned as a limitation.

-- Data collection uses a variety of qualitative methods.  however, triangulation with quantitative methods has not been done which should be mentioned as a limitation.

-- The article's aim talks about implementation of this process to all crises settings.  However, there is limited transferability to other settings which must be pointed out in the limitations section.

-- Finally the future approach must suggest full community participation instead of mere community consultation.

-- Overall, great effort.

Author Response

Dear Reviewer, 

Thank you for taking the time to review our manuscript. Please find the point-by-point response below. 

There are very few studies available from Zimbabwe and authors should be commended for taking up this study.  Handwashing is  an important behavior during humanitarian crises which is the focus of this study. The article describes the qualitative process evaluation of Wash'Em handwashing program.

Response: We appreciate the reviewer's positive feedback and recognition of the importance of studying handwashing behaviours during humanitarian crises

The introduction has been written well and is succinct.

Response: We appreciate the reviewer's positive feedback.

The design is good. It talks about coverage, fidelity, acceptability, context, etc. However, in a process evaluation quality control is an important component which the chosen process evaluation model does not fully take into account.  Perhaps this should be mentioned as a limitation.

Response: Thank you for raising this point. While our process evaluation framework did not explicitly include a 'quality control' domain, we assessed intervention fidelity throughout both the design and implementation phases. As detailed in the results, we evaluated fidelity by observing whether the Wash'Em tools were used as intended during design and whether the recommended program activities were delivered with quality and adaptations during implementation. We agree that formally auditing and providing feedback on quality is an important element we did not cover. However, by tracking fidelity we were able to highlight key deviations from intended processes and content for future improvements.

Data collection uses a variety of qualitative methods.  However, triangulation with quantitative methods has not been done which should be mentioned as a limitation.

Response: 

It was our intention to complement the qualitative data with a quantitative survey. In the limitations section of the paper we already include information about why this was not done and the limitations this created for the work: 

“Compressed timelines also resulted in us having to drop a household before and after survey which was intended to collect data on exposure to the intervention, mediating factors and behavioural outcomes (assessed through a proxy measure of whether handwashing facilities with soap and water were available. Unfortunately, this means that our understanding of intervention mediators and behavioural impact is self-reported and likely to be subject to social desirability bias.”

The article's aim talks about implementation of this process to all crises settings.  However, there is limited transferability to other settings which must be pointed out in the limitations section.

Response

We have revised the paragraph related to aims to more clearly articularly that the primary purpose is to understand contextual pathways of change for this programme in Zimbabwe and contribute to the broader aim of understanding whether Wash’Em improves the process for developing acceptable, feasible and context-appropriate hand hygiene programmes in crisis-affected settings. We recognise that this work alone cannot fulfil this broader aim. In the limitations section we have also added information about the transferability of findings. 

Finally the future approach must suggest full community participation instead of mere community consultation.

We have revised this in the discussion to focus on community participation

Response:

Overall, great effort.

Response: We appreciate the reviewer's positive feedback.

Reviewer 2 Report

Comments and Suggestions for Authors

The introduction, methods, results, and conclusion of this study were logically and well described.

However, the purpose of this study needs to be revealed more clearly at the end of the introduction.

There is a need to be more specific about the analysis method of the interview data.

Author Response

Dear Reviewer, 

Thanks for taking the time to review our manuscript. Please find the point-by-point response below. 

The introduction, methods, results, and conclusion of this study were logically and well described.

Response: We appreciate the reviewer's positive feedback.

However, the purpose of this study needs to be revealed more clearly at the end of the introduction.

Response: We have revised this paragraph on aims based on this feedback and that of other reviewers. 

There is a need to be more specific about the analysis method of the interview data.

Response: We had initially limited the detail on this give that it is already a fairly long manuscript. However we have now included details on the 6 steps of analysis that we alluded to.

Reviewer 3 Report

Comments and Suggestions for Authors

The introduction was comprehensive with a succinct explanation of the WashEm phases of the program. I would recommend providing some statistics regarding the number of deaths from inadequate hygiene in Zimbabwe and at the global level, as well.

Design/methodology

-described the socioeconomic setting and challenges in Zimbabwe

Table 2 is useful in describing the domains that were evaluated with data collection

-practical description of the methods for data collection

I would recommend including the semi-structured observation forms as an appendix in case other researchers may want to replicate this study with other populations

Results

The results were very thorough and presented a lot of "eye-opening" information such as lack of handwashing facilities, water, and soap not being accessible in locations most needed.

The interview excerpts were very useful in understanding the challenges of staff and community members

Discussion: Provided a thorough discussion of findings within the context of funding challenges, staff challenges, and humanitarian aid efforts. It is highly concerning that staff did not perceive the high rates of diarrhea to be problematic. Are there any plans to address this in the future? This can also impact the motivation of the staff to accurately implement the WashEm program.

Author Response

Dear Reviewer, 

Thank you for taking the time to review our manuscript. Please find the point-by-point response below. 

The introduction was comprehensive with a succinct explanation of the WashEm phases of the program. I would recommend providing some statistics regarding the number of deaths from inadequate hygiene in Zimbabwe and at the global level, as well.

Response: Thanks for the feedback. The authors felt that this information would be suitable in the description of the context, which is found under Section 2.1 Study sites and population demographics. We’ve edited this section to provide hygiene coverage nationally. Recent updates on the number of deaths from inadequate hygiene for Zimbabwe is not available. The authors believe the introduction sufficiently explains the link between hygiene, disease incidence and mortality. 

Design/methodology

Described the socioeconomic setting and challenges in Zimbabwe

Table 2 is useful in describing the domains that were evaluated with data collection

Response: We appreciate the reviewer's positive feedback.

Practical description of the methods for data collection

I would recommend including the semi-structured observation forms as an appendix in case other researchers may want to replicate this study with other populations

Response: The qualitative interview gudies used for the in-depth interviews are already included in the supplementary files. The document is titled “S6. Document_In depth Interview guide for implementing staff”.

Results

The results were very thorough and presented a lot of "eye-opening" information such as lack of handwashing facilities, water, and soap not being accessible in locations most needed.

Response: We appreciate the reviewer's positive feedback.

The interview excerpts were very useful in understanding the challenges of staff and community members

Response: We appreciate the reviewer's positive feedback.

Discussion: Provided a thorough discussion of findings within the context of funding challenges, staff challenges, and humanitarian aid efforts. It is highly concerning that staff did not perceive the high rates of diarrhea to be problematic. Are there any plans to address this in the future? This can also impact the motivation of the staff to accurately implement the WashEm program.

Response: We have added in a sentence in relation to this. Sadly this is not the first time we have come across this in our work. We have added references to two other papers we have worked on where this emerged as a finding. 

Reviewer 4 Report

Comments and Suggestions for Authors

Abstract:

Should include some context related to the overall takeaways of implementing the Wash’em program.

Introduction:

Have evaluations of the Wash ‘Em program been published previously?  What gap in the literature is addressed by this specific manuscript?

Methods:

Was inter-rater reliability calculated?

How were the videos and photos qualitatively analyzed?

How was the secondary analysis of operational documents and program reports conducted?  Very little context is provided within the manuscript.

Why was sampling of Wash’Em implementers designed to include a mix of genders? What did the results show (or previous literature) that might be related to these gender differences?

Additional detail related to the coding of interview and focus group data should be provided.

Results:

Who helped design the semi-structured interview forms?

How were staff trained to conduct the observations and focus groups?

In Table 4 - Do the specified quotes epitomize the overall conversations related to the issues, or are these stand-alone quotes? 

For Table 5 – Identify what the check marks and X’s mean – I am assuming check mark means recommended, and X means not recommended.

Line 379 – “Were” should be “where.”

Line 380 – “Was” should be “were.”

Line 394 – “Was” should be “were.”

Line 430 – Delete the word “were.”

Discussion:

The first paragraph should provide some type of focus – that even though the Wash’Em process was not fully implemented as intended, important conclusions were drawn related to implementing this program in a resource-scarce area...

Limitations:

Authors stated that the second round of focus groups were done in May 2023 as they concluded saturation had not been reached during the first round of focus groups. How did this impact the results?

Authors stated that the Rapid Assessments were not piloted (line 248). To what extent was this a limitation?

To what extent was the limited time allocated for data collection (Line 25) a limitation?

Line 654 – “reply” should be “rely”

Line 669 – Should read “ it is challenging not only to…”

Author Response

Dear Reviewer, 

Thanks for taking the time to review our manuscript. Please find the point-by-point response below. 

Abstract:

Should include some context related to the overall takeaways of implementing the Wash’em program.

Response: A sentence has been added on this at the end of the abstract

Introduction:

Have evaluations of the Wash ‘Em program been published previously?  What gap in the literature is addressed by this specific manuscript?

Response: This is the first evaluation of Wash’Em published in a peer-reviewed journal. In addition to this we have conducted a secondary data analysis of summary data entered into the Wash’Em Programme Designer Software. This summary data was drawn from the use of the Wash’Em Rapid Assessment Tools in 38 settings during the pandemic. This study aimed to compare data emerging from the use of the Wash’Em process during the pandemic, to understand whether commonalities in programming constraints or the determinants of handwashing behaviour existed across countries.

This process evaluation was designed to track ‘on the ground’ experiences with implementing each phase of Wash’Em in a crisis-affected setting in Zimbabwe. This was done with a view to understanding contextual pathways of change in this setting, while also contributing to the broader aim of understanding whether Wash’Em improves the process for developing acceptable, feasible and context-appropriate hand hygiene programmes in crisis-affected settings

Methods:

Was inter-rater reliability calculated? Must re-read the manuscript but not sure which part of the methods this refers to.

Response: As we used a thematic analysis approach involving collaborative coding and resolving discrepancies through consensus discussions between coders, calculating a statistical measure of inter-rater reliability was not applicable. However, undertaking collaborative coding and deliberation to resolve differences enhanced analytical rigor as themes emerged through negotiated consensus rather than reflecting any single researcher’s biases.

How were the videos and photos qualitatively analyzed?

Response: Photos and videos were discussed by the evaluation team (LSHTM, BRTI and monitoring staff from ACF) and compared to the intended stages of Wash’Em use and the intended implementation of Wash’Em designed activities.  

How was the secondary analysis of operational documents and program reports conducted?  Very little context is provided within the manuscript. 

Response: We have elaborated on the analysis of programmatic documents in the manuscript. Programmatic documents were also reviewed by the evaluation team. Through discussion, the evaluation team came to a consensus understanding about the degree of fidelity and adaptation made during implementation. This allowed links to be made between the data in the programmatic documents and the qualitative interview and observation data also collected as part of the process evaluation. 

Why was sampling of Wash’Em implementers designed to include a mix of genders? What did the results show (or previous literature) that might be related to these gender differences? 

Response: This disaggregation was based on previous hygiene related work where the presence of female/male staff assessing hygiene behaviours had a bearing on outcomes. This is now incorporated into the manuscript text.

Additional detail related to the coding of interview and focus group data should be provided.

Response: We added details on this in the manuscript text. 

Results:

Who helped design the semi-structured interview forms?

Response: The semi-structured interview forms were developed by LSHTM and reviewed by BRTI and ACF.

How were staff trained to conduct the observations and focus groups?

Response: We’ve included details on how the staff were trained. The data was collected by two very experienced qualitative researchers and one assistant trained by the researchers. 

In Table 4 - Do the specified quotes epitomize the overall conversations related to the issues, or are these stand-alone quotes?

Response: The quotes are representative of the general consensus of the conversations had in in-depth interviews and focus group discussions. 

For Table 5 – Identify what the check marks and X’s mean – I am assuming the check mark means recommended, and X means not recommended.

Response: The reviewer is correct in their assumptions and we’ve included this explanation in the manuscript text. 

Line 379 – “Were” should be “where.”

Response: This has been corrected.

Line 380 – “Was” should be “were.”

Response: This has been corrected.

Line 394 – “Was” should be “were.”

Response: This has been corrected.

Line 430 – Delete the word “were.”

Response: This has been corrected.

Discussion:

The first paragraph should provide some type of focus – that even though the Wash’Em process was not fully implemented as intended, important conclusions were drawn related to implementing this program in a resource-scarce area…

Response: This has now been added, incorporating the phrasing suggested by the reviewer. 

Limitations:

Authors stated that the second round of focus groups were done in May 2023 as they concluded saturation had not been reached during the first round of focus groups. How did this impact the results?

Response: The second round of focus groups allowed us to reach data saturation, providing fuller perspectives and enhancing confidence in the themes identified. Conducting focus groups at two timepoints also let us probe how perceptions of the intervention evolved across subgroups over several months.

Authors stated that the Rapid Assessments were not piloted (line 248). To what extent was this a limitation? 

We have added clarity to this sentence. Specifically saying that they were not piloted in this context as per the recommended Wash’Em process. The Rapid Assessment tools were piloted in 30 crises and refined accordingly as part of their development but the nuances of translation to local language mean that it is still useful to pilot each time they are used. 

To what extent was the limited time allocated for data collection (Line 25) a limitation?

Response: The limited time allocated for data collection was complicated by the fact that the population was relatively dispersed, the districts were difficult to traverse, and the teams were only able to allocate 7 hours to data collection in each site. Due to these tight time constraints only three of the Rapid Assessment were utilised, with Disease Perception and Personal Histories being omitted in both districts because AA staff felt they were less relevant to their context.

Line 654 – “reply” should be “rely”.

Response: This has been corrected. 

Line 669 – Should read “ it is challenging not only to…”

Response: This has been corrected.